# Qualitative Recognition of Primary Taste Sensation Based on Surface Electromyography

**DOI:** 10.3390/s21154994

**Published:** 2021-07-23

**Authors:** You Wang, Hengyang Wang, Huiyan Li, Asif Ullah, Ming Zhang, Han Gao, Ruifen Hu, Guang Li

**Affiliations:** State Key Laboratory of Industrial Control Technology, College of Control Science and Engineering, Zhejiang University, Hangzhou 310027, China; king_wy@zju.edu.cn (Y.W.); 11432014@zju.edu.cn (H.W.); huiyanli@zju.edu.cn (H.L.); asifkh@zju.edu.cn (A.U.); drystan@zju.edu.cn (M.Z.); gao_han@zju.edu.cn (H.G.); 0011377@zju.edu.cn (R.H.)

**Keywords:** primary tastes, taste sensation recognition, random forest, brain-computer interface (BCI), surface electromyography (sEMG)

## Abstract

Based on surface electromyography (sEMG), a novel recognition method to distinguish six types of human primary taste sensations was developed, and the recognition accuracy was 74.46%. The sEMG signals were acquired under the stimuli of no taste substance, distilled vinegar, white granulated sugar, instant coffee powder, refined salt, and Ajinomoto. Then, signals were preprocessed with the following steps: sample augments, removal of trend items, high-pass filter, and adaptive power frequency notch. Signals were classified with random forest and the classifier gave a five-fold cross-validation accuracy of 74.46%, which manifested the feasibility of the recognition task. To further improve the model performance, we explored the impact of feature dimension, electrode distribution, and subject diversity. Accordingly, we provided an optimized feature combination that reduced the number of feature types from 21 to 4, a preferable selection of electrode positions that reduced the number of channels from 6 to 4, and an analysis of the relation between subject diversity and model performance. This study provides guidance for further research on taste sensation recognition with sEMG.

## 1. Introduction

As a dominant motivation and regulator during the human feeding process, taste sensation has evolved to detect nutritive or harmful compounds in food and adjust behaviors, leading to acceptance or rejection of potential foods [1]. As a virtual simulation of taste sensation, the virtual taste has received much attention recently [2]. Many studies have focused on the approach to stimuli the tongue and provide a taste experience [3,4]. Virtual taste technology aims to provide a close-to-reality taste experience, and a key issue is a method to objectively describe human real taste sensation and evaluate the similarity between virtual taste and real taste. In the form of a wave, stimuli of visual and auditory sensation could be described by the description of the wave itself. While taste sensation is stimulated in the form of chemical substances that are difficult to describe. For most studies, the taste description was based on artificial tongues or via scales. However, for artificial tongues, which focus on describing taste substance components with sensors, describing a virtual taste sensation without real tastants is impossible [5,6,7], and scales are subjective [8]. The taste sensation description is still under investigation. To meet this need, a brain-computer interface (BCI)-based sensing system was developed herein, which could monitor the physiological variation of the subjects when they experienced different tastes.

The BCI system was employed to acquire, process, and analyze the electrophysiological or hemodynamic activity signals of the subjects [9,10]. Many studies focused on analyzing the relationship between taste stimuli and physiological signals through the use of an electroencephalogram (EEG) [11,12,13], magnetoencephalography (MEG) [13], and functional magnetic resonance imaging (fMRI) [14]. Especially, most studies used an EEG for its non-invasion and convenience [15]. The EEG has shown its potential on olfactory sensation recognition, which is similar to taste sensation recognition [16,17], and brain cognitive function detection based on multiple primary sensations [18]. In the taste sensation recognition field, an EEG could be utilized to analyze the effect of taste stimuli from food, such as flavorful creams, tomato sauce, and wine, etc. [19,20,21]. Furthermore, the taste is generally considered a combination of five primary tastes: sourness, sweetness, bitterness, saltiness, and umami [1,22]. Hence, a fundamental step of taste sensation recognition should be the qualitative recognition of those primary taste sensations. Some progress has been made based on the EEG. For instance, Abidi used an EEG for assessing sweetness and sourness and got an accuracy of 98% [23]; Andersen analyzed EEG differences under the stimuli of caloric sucrose, low-caloric aspartame, and a low-caloric mixture of aspartame and acesulfame K [24]. However, the EEG has the problem of excessive channels, information redundancy, signals capturing complexity, and data processing difficulty [10,11], while a surface electromyography (sEMG) system could solve these problems. After taste stimuli, human facial muscle movements and salivary glands activity could be recorded with non-invasive electrodes in an sEMG system [25,26,27]. sEMG reflects the electrical activity of motor units, which is combined by motor neurons and muscle fibers, and driven by the brain’s electrical activities. Therefore, sEMG is a favorable alternative to an EEG and possesses potential value in taste sensation recognition. sEMG has been widely applied in many fields such as speech recognition, gesture recognition, prosthesis control, etc. [28,29,30,31]. In the taste sensation recognition field, most research has focused on analyzing the variation of sEMG after taste stimuli. For instance, Hu analyzed sEMG of levator labii superioris under the stimuli of apple juice, Gatorade, water, soybean milk, and pickle juice [32]; Miura evaluated the effects of taste solutions, carbonation, and temperature with sEMG during the swallowing process [33]; Sato utilized sEMG to assess hedonic response during food consumption [34]. Armstrong investigated the facial EMG response of children to four tastants as a potential measurement of pleasant and unpleasant taste stimuli [35]. Horio evaluated facial expression patterns with EMG of the facial and chewing muscles, induced by eight tastants [36]. However, research on taste sensation recognition based on sEMG is still a blank.

In this study, a qualitative recognition of primary taste sensation was carried out, as shown in Figure 1. Experiments were designed and implemented to acquire sEMG of six taste sensations (no taste and five primary tastes). Five taste substances were used: distilled vinegar, white granulated sugar, instant coffee powder, refined salt, and Ajinomoto. After preprocessing and feature extraction, random forest classifiers were performed and analyzed. As a type of ensemble learning algorithm, random forest is popular for its good performance and generalization and has been widely used on the pattern recognition of sEMG [37,38,39]. Performed on a dataset containing 3054 samples with six types of labels, the random forest classifier gave a five-fold cross-validation accuracy of 74.46%, which achieved the original research purpose, i.e., developing a primary taste qualitative recognition method based on sEMG. Then we analyzed the classifier performance and simplified the model. We reduced the feature dimension from 126 to 16, by reducing the number of feature types from 21 to 4 and reducing the number of channels from 6 to 4, without apparent impact on model performance. Furthermore, we tested the model performance trained on multiple subjects.

## 2. Materials and Methods

### 2.1. Data Acquisition

Seven subjects (4 males and 3 females), aged 22 to 25 years with an average age of 23 years participated in this study with content. None of the subjects had any taste disorders and during the whole experiment period, none of the subjects had any disease, medication, or other reports about taste disorders. The research associated with human subjects was approved by the Ethics of Human and Animal Protection Committee of Zhejiang University.

The sEMG data was captured directly from facial skin with a multi-channel sEMG data recording system. Standard, non-invasive Ag/AgCl electrodes were arranged on the face of the subject, according to the position of the parotid glands and muscles, including masseters, depressor anguli oris, and depressor labii inferioris. The shape of the electrode was a circle with a diameter of 1.5 cm, and the diameter of the adhesive ring was 3.6 cm. The parotid gland is related to salivary secretion, and the muscles are related to emotional expression after taste stimuli. According to the study of Horio, the activity of masseter and depressor anguli ois could be significantly affected by taste stimuli [36], and the activity of the depressor labii inferioris could provide supplementary information about the activity of depressor anguli ois, which are also regulated by the facial nerve mandibular margin branch. Electrode distribution is shown in Figure 2. The system captured six channels of the sEMG with a sampling frequency of 1000 Hz. Channels 1 and 2 were differential electrodes and arranged on the region of the right parotid gland and masseter, while channels 5 and 6 were single electrodes on the region of the left parotid gland and masseter. Channels 1 and 5 were on the upper end of the parotid gland and masseter, while channels 2 and 6 were on the lower end. For a pair of differential electrodes, the connecting line between the two electrodes’ centers was perpendicular to the fiber’s direction of the superficial masseter. The adhesive rings’ edges of the two electrodes were tangent, which means the center-to-center distance of these two electrodes was 3.6 cm. Channel 3 was on the depressor anguli oris and channel 4 was on the depressor labii inferioris. The bias electrode and reference electrode were on the mastoid behind the left and right ears, respectively. The bias electrode is an electrode that inverts and amplifies the average common-mode signal back into the subject to significantly cancel common-mode interference and the reference electrode provides a reference potential for monopolar measurement. The position of the electrode can influence the signal [40,41], therefore, we have listed the detailed positions in Table 1. However, as face size differed for different subjects, the distances listed in the table were fine-tuned according to the muscle positions, and the amount of change in fine-tuning was fixed for the same subject.

Before the experiment, the subject was forbidden to eat for 1 h or consume irritating foods for 24 h. Each experiment contained 1 or 2 sessions, and each session included 6 trials corresponding to 6 tastes. Each experiment was carried out following 11 steps.
(1)An experiment assistant checked the sEMG acquisition system.(2)A subject with stable consciousness and a cleaned face was asked to relax and the assistant stuck electrodes on the face of the subject according to Figure 2.(3)The subject was asked to close his/her eyes, think about nothing, and not control their facial expressions on purpose. Then the assistant started a device to capture the sEMG of the experiment category “No taste” for 12 s.(4)The subject gargled five times, taking 20 mL purified water each time.(5)The subject relaxed for 2 min.(6)The subject closed his/her eyes, then the assistant prepared one level spoon of instant coffee powder (UCC117) and inverted it on the tip of the subject’s tongue. The edge of the spoon was a circle with a radius of 11 cm and the volume of the spoon was 2 mL. The position of stimulus coverage is shown in Figure 3.(7)The subject backed his/her tongue, closed his/her mouth, and kept the tongue still. The subject was forbidden to control their facial expressions on purpose or think about anything but the taste experience. Then the assistant started the device to get an sEMG of the bitter taste for 12 s.(8)Repeat steps 4 and 5.(9)Repeat the steps from 6 to 8 and replace the instant coffee powder with white granulated sugar (Taigu, sucrose ratio was not less than 99.6%), refined salt (Zhongyan, sodium chloride ratio was not less than 98.5%), Ajinomoto (Xihu, sodium glutamate ratio was not less than 99.9%), and distilled vinegar (Hengshun, the total acid content was not less than 5 g per 100 mL) for sweet, salty, umami, and sour taste, respectively.(10)The subject took a break for 10 min.(11)If there was another session, repeat steps from 3 to 10.

The order of different tastes was decided randomly. Thirty-nine experiments were carried out, comprising 65 sessions.

### 2.2. Preprocessing

To augment samples, data acquired by the system were cut into samples by a sliding window. Although granular materials are not liquid, they melted during each trial. The melting process would continue to the end of the trial, as confirmed by residual materials while gargling. Therefore, sEMG data acquired by the system reflected motor unit activities under continuous taste stimuli. Data of each trial were cut into samples with a sample length of 1 s and a step length of 0.25 s.

### 2.3. Signals in Samples Were Filtered

The sEMG signal captured on facial skin was affected by baseline drift, and power frequency interference. Baseline drift, caused by the change in facial muscle tension and zero drift of the device, was removed as a trend term of a quartic polynomial. The following step was a fourth-order high-pass Butterworth filter at a cutoff frequency of 10 Hz. To remove power frequency interference, an adaptive notch filter was applied to signals. The filter can modify signals with a high amplitude at 50 Hz and harmonics without destroying most normal signals. An example of preprocessing is shown in Figure 4. For several samples, the power frequency interference was so strong that our adaptive notch filter could not filter it without signal distortion, and we had to remove these samples. Feature extraction.

Sample signal spectrums indicated that the energy of the sEMG was concentrated from 10 Hz to 100 Hz. It is reasonable that a circular electrode would introduce a low-pass filtering effect and an sEMG signal captured with such a large electrode (diameter of 1.5 cm) would result in a large filtering effect [41,42]. For each channel, the spectrum was divided into 13 intervals. From 10 Hz to 100 Hz, the interval length was 10 Hz, while from 100 Hz to 500 Hz, the interval length was 100 Hz. After detrending and filtering during the preprocess, the spectrum below 10 Hz contained little information anymore. Therefore, the interval between 0 and 10 Hz was not included. An example of the spectrum and interval division is shown in Figure 5. For each interval, the integral value of amplitude was calculated as a feature. To further reflect the energy distribution of the spectrum, three other features were calculated, including frequency centroid (FC), root mean square frequency (RMSF), and root var frequency (RVF).

Five time-domain features were also taken into consideration, including root mean square (RMS), zero-crossing rate (ZCR), mean absolute value (MAV), kurtosis (Ku), and skewness (Kw).

For each channel, all 21 types of features were calculated according to the following equations:(1)$$F\left[n\right]=\frac{1}{10}\sum_{i=n\times 10}^{\left(n+1\right)\times 10-1}f\left(i\right),\;when\;1\leq n\leq 9\;$$
(2)$$F\left[n\right]=\frac{1}{100}\sum_{i=\left(n-9\right)\times 100}^{\left(n-8\right)\times 100-1}f\left(i\right),\;when\;10\leq n\leq 13\;$$
(3)$$F\left[14\right]=\frac{\sum_{i=1}^{L/2}f\left(i\right)\times \left(i\times f_{s}/L\right)}{\sum_{i=1}^{L/2}f\left(i\right)}\;$$
(4)$$F\left[15\right]=\sqrt{\frac{\sum_{i=1}^{L/2}f\left(i\right)\times {\left(i\times f_{s}/L\right)}^{2}}{\sum_{i=1}^{L/2}f\left(i\right)}}\;$$
(5)$$F\left[16\right]=\sqrt{\frac{\sum_{i=1}^{L/2}f\left(i\right)\times {\left(i\times f_{s}/L-F\left[14\right]\right)}^{2}}{\sum_{i=1}^{L/2}f\left(i\right)}}\;$$
(6)$$F\left[17\right]=\sqrt{\frac{\sum_{i=1}^{L}x_{i}{}^{2}}{L}}\;$$
(7)$$F\left[18\right]=\sum_{i=1}^{L-1}1,\;when\;x_{i}x_{i+1}<0\;$$
(8)$$F\left[19\right]=\frac{\sum_{i=1}^{L}\left|x_{i}\right|}{L}\;$$
(9)$$F\left[20\right]=\frac{1}{L}\sum_{i}^{L}{\left(\frac{x_{i}-\mu }{\sigma }\right)}^{4}\;$$
(10)$$F\left[21\right]=\frac{1}{L}\sum_{i}^{L}{\left(\frac{x_{i}-\mu }{\sigma }\right)}^{3}$$

*F*[*n*] is the nth feature. *f*(*i*) is the spectrum amplitude of the *i*th point under the frequency domain. *L* is the length of a sample, and in this case, *L* is 1000. *f_s_* is the sampling frequency, and in this case, *f_s_* is 1000. *x_i_* is the value of the *i*th data point under the time domain. *μ* and *σ* are the mean value and standard deviation of all data points.

All 21 dimensions features for each channel are listed in Table 2. Since we had 6 channels and 21 types of features for each channel, we extracted a feature set containing 126 features.

### 2.4. Classification

Random forest, an ensemble learning method based on the decision tree, is a popular learning method of pattern recognition for its high performance and robustness. We performed the random forest algorithm on a dataset containing all samples of subject 1 to manifest the feasibility of the recognition task. To simplify the model, we selected features based on model performance on datasets with different features and feature importance, provided by the random forest algorithm. After the preferable feature selection was chosen, we performed the algorithm on different datasets with all 63 possible channel combinations of 6 channels and selected the best channel combination. Then, to analyze the impact of subject diversity, we constructed 127 datasets with all possible combinations of 7 subjects, performed the random forest algorithm on them, and compared the average accuracy for models trained on different numbers of subjects. For all datasets mentioned above, the model performance was evaluated with the accuracy of 5-folds cross-validation. Different samples augmented from the same experimental session were assigned to the same fold.

## 3. Results and Discussion

### 3.1. Classification Result

We utilized a dataset name “Dataset 01” to verify the feasibility of primary taste sensation qualitative recognizing with sEMG. Dataset 01 was comprised of all 3054 samples from subject 1. The numbers of samples with the labels “None”, “Sour”, “Sweet”, “Bitter”, “Salty”, and “Umami” were 532, 492, 522, 488, 485, and 536, respectively. Random forest was performed on Dataset 01, giving a five-fold cross-validation accuracy of 74.46%. The confusion matrix is shown in Figure 6. Accuracies for all six types of tastes were higher than the chance level, confirming that the method is feasible and effective for all kinds of primary tastes.

### 3.2. Feature Selection

After random forest was performed on Dataset 01, the algorithm could provide feature importance for each feature, which was also known as the Gini importance, and computed as normalized total reduction of the criterion brought by the feature. We got 21 types of features and for the No.i feature type, the feature type score G_i_ could be calculated as the sum of G_ij_ for j from 1 to 6, where G_ij_ was the feature importance for the No.i feature for the channel j.

To simplify the model, we reconstructed Dataset 01 by changing the feature of samples. For each new dataset, we changed the feature combination by only adopting part of the feature types with the highest score, and the number of feature types we adopted was marked as NF. We constructed new datasets with NF = 1, 2, …, 21 and performed random forest models on the new datasets. The relation between NF and the model accuracy Acc_NF_ is shown in Figure 7.

When NF increased to 4, the curve reached the inflection point and the model accuracy was 75.74%, which was even higher than the accuracy of the model performed on Dataset 01. The result indicated that we could fulfill the taste sensation recognition task only with the four best types of features, without impact on the model performance. Sorted from most to least feature type score, the four best features were average spectrum amplitude between 300 Hz and 400 Hz, average spectrum amplitude between 200 Hz and 300 Hz, average spectrum amplitude between 100 Hz and 200 Hz, and RVF. Although most of the energy was concentrated in the range below 100 Hz, the spectrum amplitude of the range above 100 Hz showed more differences for samples with different labels. RVF was an overall evaluation of energy distribution in the spectrum, and the difference of RVF might also come from the variation of energy in the range between 100 Hz and 400 Hz. The new version of Dataset 01, with the new feature set, which contains the four best feature types, was fixed for further steps and named “Dataset 02”.

### 3.3. Channel Combination Selection

To simplify the model and analyze the preferable electrode positions, we performed a random forest on new datasets with different channel combinations. We got six channels, and there could be 15 possible combinations of these channels. We reconstructed a new dataset for each possible combination and performed a random forest on the dataset. For a channel combination, the number of channels was marked as NC. When NC was 1, the model was performed on a dataset with only one channel, whose result could be an evaluation of the information richness of the channel. The accuracies of these models were 49.00%, 55.20%, 43.20%, 41.30%, 41.60%, and 51.30%, sorted by channel number from 1 to 6. With almost the same position of electrodes, channel 2 outperformed channel 6, and channel 1 outperformed channel 5, indicating that differential electrodes showed better results, probably due to their anti-inference ability. With the same electrode type, channel 2 outperformed channel 1, and channel 6 outperformed channel 5, indicating that the lower end was a preferable position for the electrode to capture information from the parotid gland and masseter.

When NC equaled 3, the highest accuracy came from the combination of 2/3/6 (73.40%). Although channel 1 got much higher accuracy than channels 3 and 4, combination 2/3/6 and 2/4/6 (70.30%) both showed higher accuracy than 1/2/6 (69.40%). It was reasonable that the electrodes for channels 1 and 2 were stuck on the region of the parotid gland and masseter on the same side and signals of channels 1 and 2 had a high correlation, therefore, when combined with channel 2, channel 1 contained considerable redundant information.

Acc_i-best_ and Acc_i-mean_ were the best accuracy and the mean accuracy, respectively, of the models performed on datasets whose NC equalled i. The relations between NC and Acc_NC-best_ or Acc_NC-mean_ are shown in Figure 8. When NC increased to 4, the curve reached the inflection point and the model accuracy was 75.41%, which was almost the same as the accuracy of the model performed on Dataset 02. The best combination of four channels was 2/3/4/6, and the contributions from these 4 channels were enough for the recognition task. It was reasonable that these four channels have provided the information of parotid glands and masseters on both sides, depressor anguli oris, and depressor labii inferioris, while most of the information from the other two channels were redundant. Therefore, the new channel combination 2/3/4/6 was fixed for further steps.

For each sample and each channel among channels 3, 4, and 6, the sum of the spectrum amplitude between 100 Hz and 400 Hz was calculated marked as S_i_, where i means the index of the channel. For a type of primary taste, the average value of S_3_ for all samples with this primary taste label represented the activity of the depressor anguli oris under this type of taste stimuli. The spectrum amplitude of frequency below 100 Hz or above 400 Hz was not included because these features did not show potential for pattern recognition, which means the differences of tastes were not typical. Similarly, the average value of S_4_ and S_6_ represented the activity of the depressor labii inferioris and masseter respectively. Instead of channel 2, channel 6 was chosen because a signal without differential operation could reflect signal strength more intuitively. For six tastes and three selected channels, 18 S values were calculated. Then we linearly mapped the S values to grayscales, with the smallest S value mapped to 0 (grayscale of the black) and the biggest S value mapped to 255 (grayscale of the white). After the mapping, for each muscle under the stimuli of each taste, we could get a muscle activity represented in grayscale. The muscle activities are shown together in Figure 9. When the grayscale of the shadow around the muscle s smaller (the color is closer to the black) it represented that the activity of the muscle is stronger. Sweet and umami stimuli showed a similar activity of the depressor anguli oris to no taste stimuli and stronger activity of the other muscles. It could be speculated that the facial expression under sweet and umami stimuli was similar to that of no stimuli, except for a slight reaction of the cheek and underlip. It was reasonable that under stimuli of taste providing a positive hedonic value, the subject showed less facial expression than that of negative hedonic value. The muscle activity of salty and bitter stimuli represented a more apparent reaction of facial expressions. The activity of sour stimuli was much higher than other tastes. The result represented that the underlip and chin would show a much more apparent reaction to sour stimuli, which presented as apparent tremors, as observed during the experiment.

### 3.4. Model Performance on Different Subjects

After Section 3.2 and Section 3.3, we got a new feature selection and channel combination for the model. Then we tested model performance on datasets with samples from other subjects or multiple subjects. The dataset i was comprised of all samples from subject i, with the fixed feature selection in Section 3.2 and channel combination in Section 3.3. For all seven subjects, subject ID (i), dataset (dataset i) size, and accuracy of the random forest model performed on the subject’s dataset (dataset i) are shown in Table 3.

All of the models performed on a single subject dataset gave an accuracy higher than the chance level, and the average accuracy for these seven models was 56.83%. The result indicated that the method was feasible for all subjects in this study. Performed on a dataset containing all 16,257 samples from all seven subjects, a random forest model gave an accuracy of 50.02%, indicating that the method was feasible on datasets of multiple subjects.

There should be 127 possible combinations of seven subjects. For each combination, we constructed a dataset with all samples from subjects in the combination and performed random forest on the dataset. For such a dataset, the number of subjects was marked as NS. Acc_i-mean_ was the mean accuracy of the models performed on datasets whose NS equaled i. The relation between NS and Acc_NS-mean_ is shown in Figure 10. For our limited subjects, the model performance decreased with the increase of subject diversity. The stable level of accuracy could not be confirmed by our limited subjects, and still needs further research in the future.

## 4. Conclusions

In this study, the feasibility of taste sense qualitative recognition based on sEMG was verified. An experimental protocol was designed to acquire an sEMG under different taste stimuli, after which data was preprocessed and classified with a random forest classifier. The model gave a five-fold cross-validation accuracy of 74.46%, which manifested the feasibility of this recognition task.

To simplify the model, we reduced the feature dimension from 126 to 16 by reducing the number of feature types from 21 to 4 and reducing the number of channels from 6 to 4, without an apparent effect on model performance. When reducing the number of feature types, we analyzed which features made the most contributions and found that the spectrum of the range between 100 Hz and 400 Hz showed more energy differences. When reducing the number of channels, we preliminarily confirmed differential electrodes were better than single electrodes, the lower end was better than the upper end for electrodes acquiring the information of the parotid gland and the masseter, and signals from two different electrode positions on the parotid showed a high correlation.

Furthermore, we tested model performance on datasets with samples from different subjects. The recognition method was feasible for every single subject as well as multiple subjects. Judging from the limited subjects we used so far, the model performance decreased with the increase of subject diversity.

The main research target of this study was to propose a novel method for the qualitative recognition of primary taste sensation. The study proved the feasibility of the task, giving a valid approach to acquire, describe, and recognize taste sensations with sEMG. This method provides guidance for the potential development of a standard and precise description of taste sensations.

However, our study still shows many limitations. Firstly, we only carried out the recognition task in a qualitative way. More precise, quantitative research is needed in our further study. Secondly, a circular electrode would introduce a low-pass filtering effect with a transfer function related to electrode size, and for a pair of differential electrodes, the final transfer function would also be affected by the spatial filtering effect related to interelectrode distance [42]. Moreover, the spatial relationship between muscle fiber direction and the direction of differential electrode pairs would influence the sEMG signal. In addition, there are still some alternative electrode positions related to other facial muscles [36]. Therefore, the effect of electrode size, interelectrode distance, and electrode location need to be verified since they have an impact on the features of the sEMG signal. Finally, the number of subjects and samples were not enough, and more subjects and an adequate dataset might help better analyze the effect of subject diversity on model performance.

## Figures and Tables

**Figure 1 sensors-21-04994-f001:**
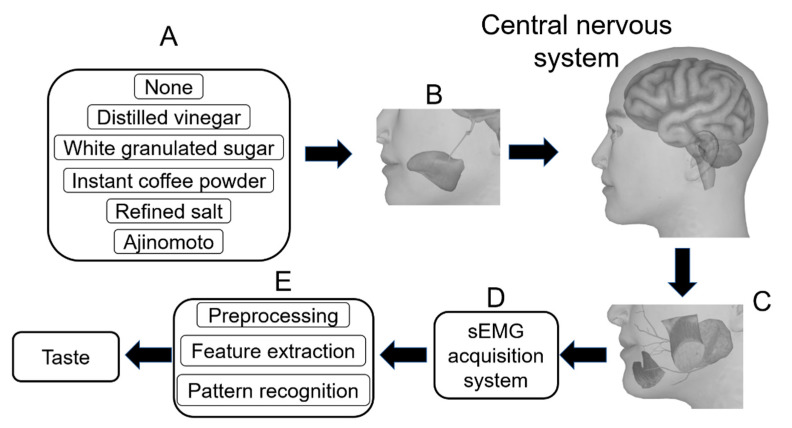
Schematic diagram of the study. (**A**) We used taste substances to provide 6 types of taste. (**B**) The tongue was stimulated by taste substance and taste information was transmitted to the central nervous system via the lingual nerve, mandibular nerve, etc. (**C**) Information was transmitted to muscles (masseters, depressor anguli oris, and depressor labii inferioris) and glands (parotid glands) via the facial nerve. (**D**) The activity of muscles and glands was recorded as sEMG by an acquisition system. (**E**) The sEMG signals were preprocessed, feature extracted, and pattern recognized.

**Figure 2 sensors-21-04994-f002:**
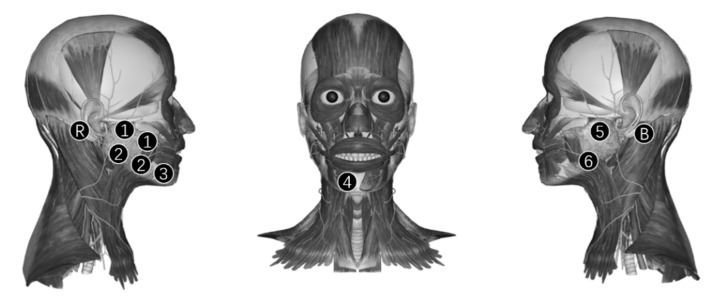
Electrodes position distribution. Circle with a number was the position of electrodes for 6 channels, and the number was the channel index. Electrodes for channels 1 and 2 were differential electrode pairs. Circle with the letters ‘B’ and ‘R’ were the positions of the bias electrode and reference electrode. Electrodes for channels 1, 2, 5, and 6 were arranged on masseters, the electrode for channel 3 was arranged on depressor anguli oris, and the electrode for channel 4 was arranged on depressor labii inferioris. In addition, the bias electrode and reference electrode were arranged on the left and right mastoid, respectively.

**Figure 3 sensors-21-04994-f003:**
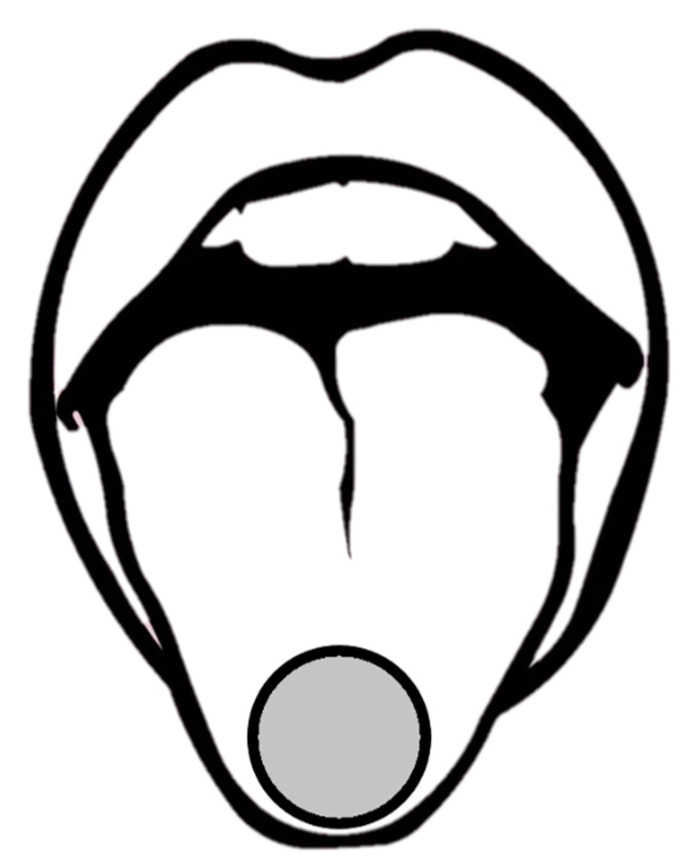
The position of stimulus coverage.

**Figure 4 sensors-21-04994-f004:**
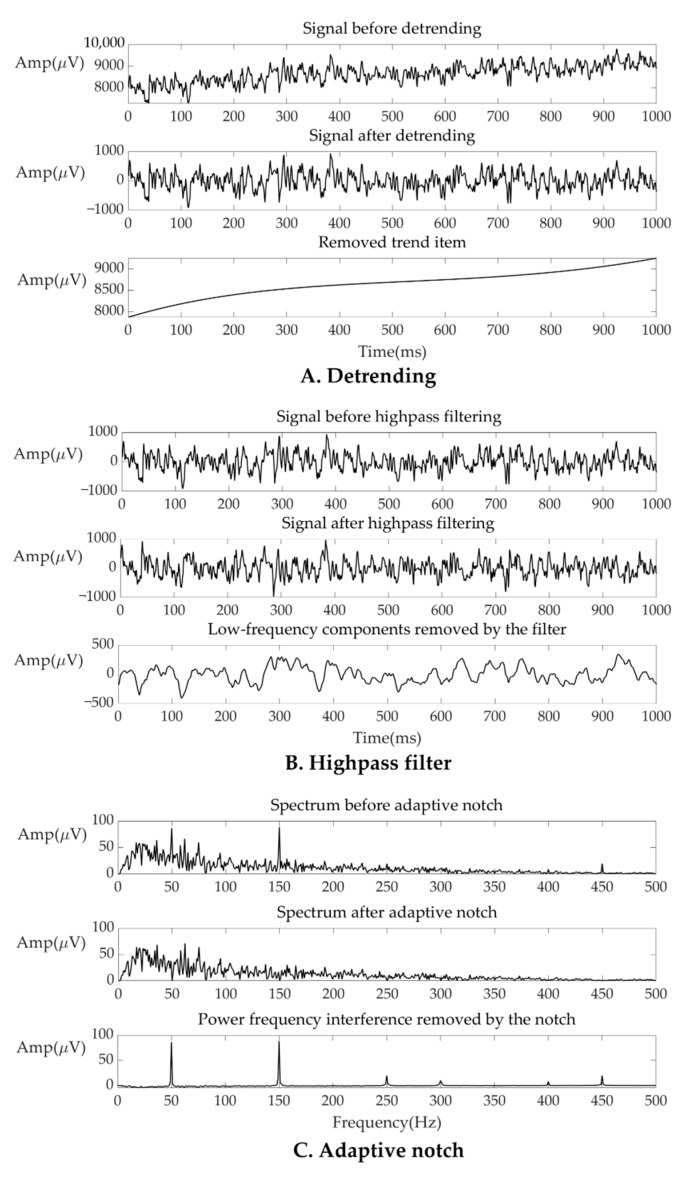
An example of the preprocessing result. (**A**) The signals before/after detrending and removed trend item (**B**) The signals before/after high-pass filter and the low-frequency components removed by the filter. (**C**) The spectrum of signal before/after adaptive notch and the spectrum of the power frequency interference removed by the notch.

**Figure 5 sensors-21-04994-f005:**
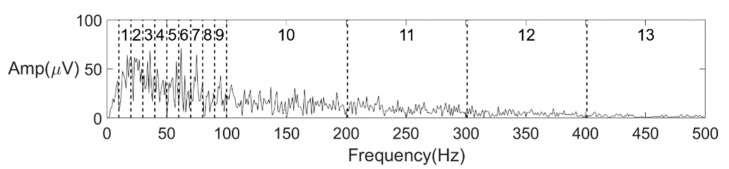
Interval division for features 1–13.

**Figure 6 sensors-21-04994-f006:**
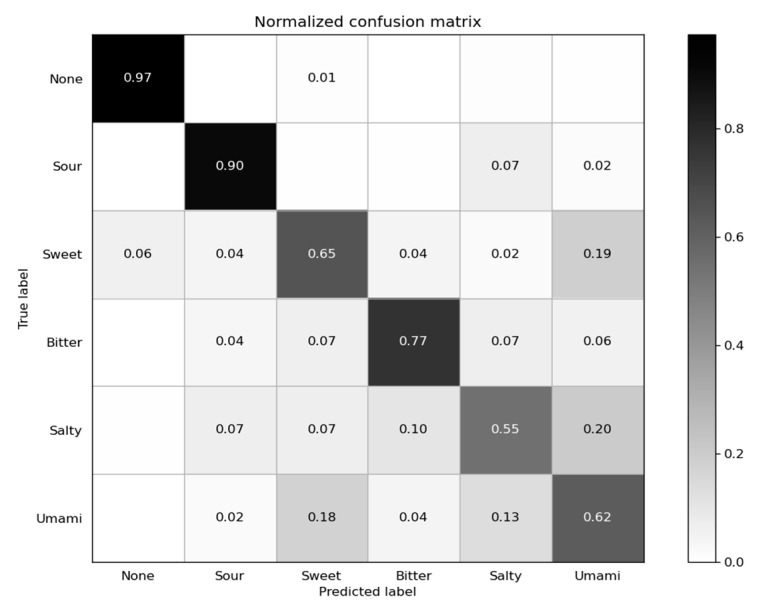
Confusion matrix on Dataset 01. Number in row i, column j is the ratio of N_ij_ (number of samples classified as label j with true label i) to N_i_ (number of samples with true label i), and values smaller than 0.01 are ignored.

**Figure 7 sensors-21-04994-f007:**
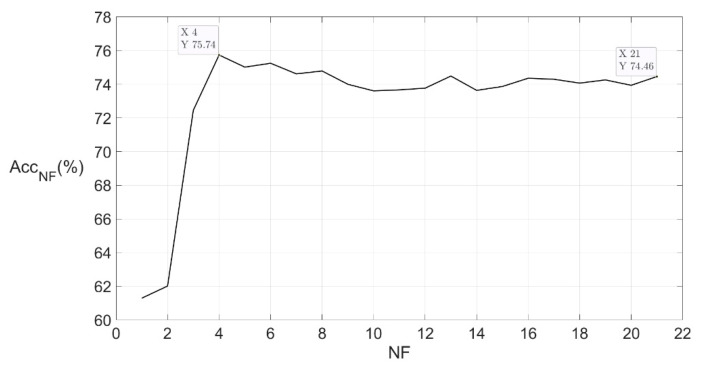
Relation between NF and Acc_NF_. NF is the number of feature types. After performing the random forest algorithm on a dataset of all features, a score for each feature type was provided. By adopting NF types of features with the highest score, we constructed a new dataset and performed a random forest on it, and the accuracy of the model was marked as Acc_NF_. The figure shows the trend of Acc_NF_ change as the NF changed.

**Figure 8 sensors-21-04994-f008:**
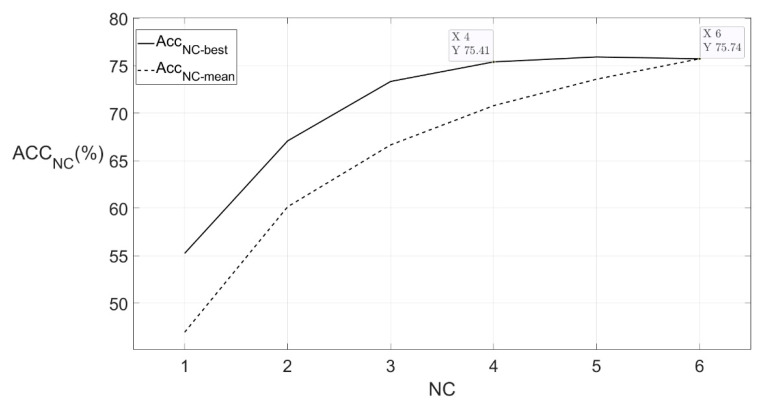
Relation between Acc_NC-best_, Acc_NC-mean,_ and NC. NC is the number of channels. For each NC, we listed all possible channel combinations of NC channels and constructed a new dataset for each combination by only adopting features from channels in the combination. For each combination, after performing the random forest algorithm, we could get an accuracy. The highest one among the accuracies was marked as Acc_NC-best_ and the mean value of the accuracies was marked as Acc_NC-mean_. The figure shows the trend of Acc_NC-best_ and Acc_NC-mean_ changed as the NC changed.

**Figure 9 sensors-21-04994-f009:**
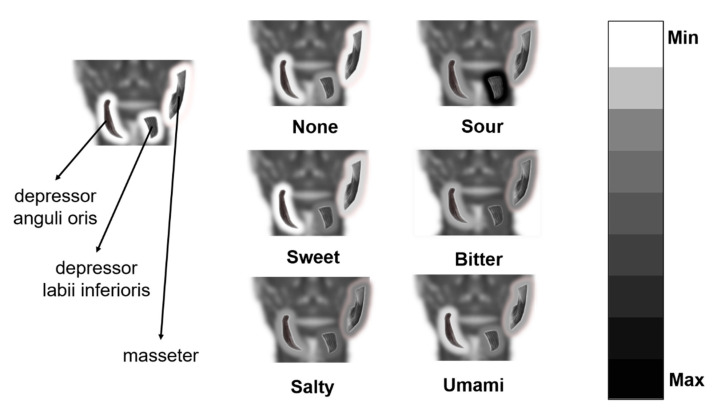
The activities of muscles under different taste stimuli. For depressor anguli oris, depressor labii inferioris, and the masseter, if the grayscale of the shadow around the muscle is smaller (the color is closer to the black) it represented the activity of the muscle is stronger under corresponding taste stimuli.

**Figure 10 sensors-21-04994-f010:**
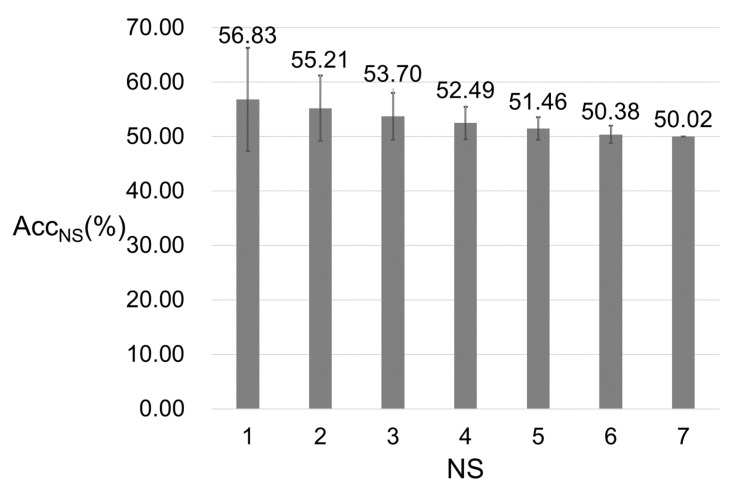
Relation between Acc_NS-mean_ and NS. NS is the number of subjects. For each NS, we listed all possible subject combinations of NC subjects and constructed a new dataset for each combination by combining datasets of subjects in the combination. For each combination, after performing the random forest algorithm, we could get an accuracy. The highest one among the accuracies was marked as Acc_NS_. The figure shows the trend of Acc_NS_ changed as the NC changed.

**Table 1 sensors-21-04994-t001:** Detailed electrode positions.

Electrode Index	Electrode Type	Muscle	Detailed Position *
1	Differential electrode pairs	Left masseter	The lower end of the right ear cartilage forward 3 cm
2	Differential electrode pairs	Left masseter	Parallel to and below electrodes for channel 1 and close to the edge of the jaw
3	Single electrode	Left depressor anguli oris	The lower right of the right corner of the mouth
4	Single electrode	Left depressor labii inferioris	Below the lower lip with the whole electrode on the right of the middle line of the face
5	Single electrode	Right masseter	The lower end of the left ear cartilage forward 3 cm
6	Single electrode	Right masseter	Below the electrode for channel 5 and close to the edge of the jaw
B	Bias electrode	Left mastoid	The protuberance behind the left ear
R	Reference electrode	Right mastoid	The protuberance behind the right ear

* The detailed position for a pair of differential electrodes was the tangent point of two circular electrodes’ adhesive rings, while the detailed position for a single electrode was the center of the circular electrode.

**Table 2 sensors-21-04994-t002:** Name, type, and the dimension of each feature.

Feature Name	Feature Type	Feature Dimension
Spectrum average amplitude	Frequency-domain	13
FC	Frequency-domain	1
RMSF	Frequency-domain	1
RVF	Frequency-domain	1
RMS	Time-domain	1
ZCR	Time-domain	1
MAV	Time-domain	1
Ku	Time-domain	1
Sk	Time-domain	1

**Table 3 sensors-21-04994-t003:** Subject ID, number of samples, and accuracy of the random forest for datasets with samples from one subject.

Subject ID	Number of All Samples of the Dataset	Accuracy of Random Forest on the Dataset (%)
1	3054	75.41
2	3085	61.49
3	2098	55.10
4	1474	48.78
5	2130	52.11
6	2912	57.07
7	1504	47.87

## Data Availability

Not applicable.

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
