# Peer review of "Qualitative Recognition of Primary Taste Sensation Based on Surface Electromyography"

_sensors, 2021, doi:10.3390/s21154994_

Round 1

Reviewer 1 Report

General:

In this study, the authors developed a novel machine learning method to distinguish primary taste sensations using surface electromyography (sEMG) signals.

This issue warrants investigation, and this is probably the first study reporting machine learning for taste sensation discrimination based on sEMG signals; hence, the data will have a significant impact on the literature. The manuscript is written clearly and concisely.

However, there are some problems in the current manuscript.

- The literature review is not adequate. Because some previous studies have tested taste sensation using sEMG (e.g., Armstrong et al., 2007: Chemical Senses, 32, 611-621; Horio, 2003: Perceptual and Motor Skills, 97, 289-298), I recommend that the authors cite these studies in the Introduction.

- The rationale for electrode placement is unclear. Although the authors described that electrodes were arranged on the muscles “related to emotional expression after taste stimuli” (p. 3), what evidence shows that the masseters, depressor anguli oris, and depressor labii inferioris are related to emotional expressions for taste stimuli? In addition, it is unclear whether these muscles were expected to reflect emotional expressions or taste sensation.

- Can the authors interpret sEMG patterns? Some readers may be interested in facial expressions in response to 6 human primary taste sensations.

Minor:

- Abbreviations (e.g., NF) should be defined when first mentioned.

Reviewer 2 Report

Thank you for the opportunity to review this manuscript. 
The main research target of this study was to propose a novel method for qualitative recognition of primary taste sensation. The study proved the feasibility of the task, giving a valid approach to acquire, describe and recognize taste sensations with sEMG. This method provides guidance for the potential development of a standard and precise description of taste sensations. 
Introduction -  is clearly 
Material and methods 
1) Please describe the participants in more detail (age, gender, medical conditions, medications, etc.)
2) Please explain on what basis it was found: None of the subjects had any taste disorders.
Preprocessing - is adequately described.
Feature extraction - Why is the lower sample signal spectrum below 10 Hz not included?
Results and discussion - is clearly 
Conclusion - I propose to list the limitations in a separate paragraph.

Round 2

Reviewer 1 Report

Great improvements have been made in the revised manuscript. The authors have addressed all points I raised.

Author Response

Thank you for your careful work and it is really helpful to make our article better.